# Why space could be quantised on a different scale to matter

**Matthew J. Lake**⋆

**1** School of Physics, Sun Yat-Sen University, Guangzhou 510275, China
**2** Frankfurt Institute for Advanced Studies (FIAS),
Ruth-Moufang-Str. 1, 60438 Frankfurt am Main, Germany
**3** Department of Physics, Babeş-Bolyai University,
Mihail Kogălniceanu Street 1, 400084 Cluj-Napoca, Transylvania, Romania
**4** National Astronomical Research Institute of Thailand (NARIT),
260 Moo 4, T. Donkaew, A. Maerim, Chiang Mai 50180, Thailand
**5** Research Center for Quantum Technology (RCQT),
Faculty of Science, Chiang Mai University, Chiang Mai, 50200, Thailand

⋆ lake@fias.uni-frankfurt.de

*4th International Conference on Holography,
String Theory and Discrete Approach
Hanoi, Vietnam, 2020*

## Abstract

**The scale of quantum mechanical effects in matter is set by Planck's constant, $\hbar$. This represents the quantisation scale for material objects. In this article, we present a simple argument why the quantisation scale for space, and hence for gravity, may not be equal to $\hbar$. Indeed, assuming a single quantisation scale for both matter and geometry leads to the 'worst prediction in physics', namely, the huge difference between the observed and predicted vacuum energies. Conversely, assuming a different quantum of action for geometry, $\beta \ll \hbar$, allows us to recover the observed density of the Universe. Thus, by measuring its present-day expansion, we may in principle determine, empirically, the scale at which the geometric degrees of freedom should be quantised.**

## 1 Wave–particle duality and $\hbar$

Classical mechanics is deterministic [1]. If its initial conditions are known, the probability of finding a particle at a given point on its trajectory, at the appropriate time $t$, is 100%. The corresponding state is described by a delta function, $\delta^3(\mathbf{x}-\mathbf{x}')$, with dimensions of (length)$^{-3}$. This is the probability density of the particle located at $\mathbf{x} = \mathbf{x}'$.

In quantum mechanics (QM), probability amplitudes are fundamental. Position eigenstates, $|\mathbf{x}\rangle$, are the rigged basis vectors of an abstract Hilbert space, where $\langle\mathbf{x}|\mathbf{x}'\rangle = \delta^3(\mathbf{x}-\mathbf{x}')$. These have dimensions of (length)$^{-3/2}$ and more general states may be constructed by the principle of quantum superposition [2]. The resulting wave function, $\psi(\mathbf{x})$, represents the probability amplitude for finding the particle at each point in space, and the corresponding probability density is $|\psi(\mathbf{x})|^2$ [3].

Since $\psi(\mathbf{x})$ can also be decomposed as a superposition of plane waves, $e^{i\mathbf{k}.\mathbf{x}}$, an immediate consequence is the uncertainty principle $\Delta_\psi x^i \Delta_\psi k_j \geq (1/2)\delta^i{}_j$, where $i,j \in \{1,2,3\}$ label orthogonal Cartesian axes. This is a purely mathematical property of $\psi$ that follows from elementary results of functional analysis [4]. In canonical QM, we relate the particle momentum $\mathbf{p}$ to the wave number $\mathbf{k}$ via Planck's constant, following the proposal of de Broglie, $\mathbf{p} = \hbar\mathbf{k}$. It follows that

$$\Delta_\psi x^i \Delta_\psi p_j \geq (\hbar/2)\delta^i{}_j. \tag{1}$$

This is the familiar Heisenberg uncertainty principle (HUP). We stress that the HUP is a consequence of two distinct physical assumptions:

1. the principle of quantum superposition, and

2. the assumption that $\hbar$ determines the *scale* of wave–particle duality. [1]

Let us also clarify the meaning of the word 'particle'. We stress that canonical QM treats all particles as point-like, so that eigenstates with zero position uncertainty may be realised, at least formally. However, gravitational effects are expected to modify the HUP by introducing a minimal length, $\Delta x > 0$ [6,7]. Next, we discuss how this relates to theoretical predictions of the vacuum energy.

## 2 Minimal length and the vacuum energy

In canonical QM, the background space is fixed and classical. Individual points are sharply defined and the distances between them can be determined with arbitrary precision [8]. By contrast, thought experiments in the quantum gravity regime suggest the existence of a minimum resolvable length scale of the order of the Planck length, $\Delta x \simeq l_{\rm Pl}$, where $l_{\rm Pl} = \sqrt{\hbar G/c^3} \simeq 10^{-33}$ cm [6]. Below this, the classical concept of length loses meaning, so that perfectly sharp spacetime points cannot exist [7].

This motivates us to take $l_{\rm Pl}$ as the UV cut off for vacuum field modes, but doing so yields the so-called 'worst prediction in physics' [9], namely, the prediction of a Planck-scale vacuum density:

$$\rho_{\rm vac} \simeq \frac{\hbar}{c}\int_{k_{\rm dS}}^{k_{\rm Pl}} \sqrt{k^2 + \left(\frac{mc}{\hbar}\right)^2}\, {\rm d}^3k \simeq \rho_{\rm Pl} = \frac{c^5}{\hbar G^2} \simeq 10^{93}\,{\rm g.cm}^{-3}. \tag{2}$$

Unfortunately, the observed vacuum density is more than 120 orders of magnitude lower,

$$\rho_{\rm vac} \simeq \rho_\Lambda = \frac{\Lambda c^2}{8\pi G} \simeq 10^{-30}\,{\rm g.cm}^{-3}. \tag{3}$$

---

[1]Note that these assumptions are consistent with Poincaré invariance, and, hence, with Galilean invariance in the non-relativisitc limit of canonical QM, if and only if $\mathbf{p} \propto \mathbf{k}$ and $E \propto \omega$ [5]. Ultimately, it is the constant of proportionality in these relations that determines the length and momentum (energy) scales at which quantum effects become important. The 'quantisation scale' of any system is, therefore, an action scale, which must be determined empirically. For canonical quantum particles, this scale is $\hbar = 1.05 \times 10^{-34}$ J s.

In Eq. (2), the mass scale $m \ll m_{\rm Pl} = \hbar/(l_{\rm Pl}c) \simeq 10^{-5}$ g is set by the Standard Model of particle physics [10] and the limits of integration are $k_{\rm Pl} = 2\pi/l_{\rm Pl}$, $k_{\rm dS} = 2\pi/l_{\rm dS}$, where $l_{\rm dS} = \sqrt{3/\Lambda}$ is the de Sitter length. This is comparable to the present day radius of the Universe, $r_{\rm U} \simeq 10^{28}$ cm, which may be expressed in terms of the cosmological constant, $\Lambda \simeq 10^{-56}$ cm$^{-2}$ [11].

More detailed calculations alleviate this discrepancy [12], but our naive calculation highlights the problem of treating $l_{\rm Pl}$ and $m_{\rm Pl}$ as interchangeable cutoffs. We now discuss an alternative way to obtain a minimum length of order $l_{\rm Pl}$ without generating unfeasibly high energies.

## 3   Wave–point duality and $\beta \neq \hbar$

Clearly, one way to implement a minimum length is to discretise the geometry, as in loop quantum gravity and related approaches [13]. However, in general, quantisation is *not* discretisation [14]. The key feature of quantum gravity is that it must allow us to assign a quantum state to the background, giving rise to geometric superpositions, and, therefore, superposed gravitational field states [15]. The associated spectrum may be discrete or continuous, finite or infinite.

But how to assign a quantum state to space itself? One possible, but simple, answer is that we must begin by assigning a quantum state to each *point* in the classical background. Individual points can then be mapped to superpositions of points, which results in the unique classical geometry being mapped to a superposition of geometries, as required [16]. In effect, we may apply the quantisation procedure point-wise, and, in the process, eliminate the concept of a classical point from our description of physical reality.

This can be achieved by first associating a rigged basis vector, i.e., a ket $|\mathbf{x}\rangle$ with each coordinate '$\mathbf{x}$'. We then note that $\langle \mathbf{x}|\mathbf{x}'\rangle = \delta^3(\mathbf{x} - \mathbf{x}')$ is obtained as the zero-width limit of a probability Gaussian distribution, $|g(\mathbf{x} - \mathbf{x}')|^2$, with standard deviation $\Delta_g x$. Taking $\Delta_g x > 0$ therefore 'smears' sharp spatial points over volumes of order $\sim (\Delta_g x)^3$, giving rise to a minimum observable length scale [16]. Motivated by thought experiments [6], we set $\Delta_g x \simeq l_{\rm Pl}$.

Since $g$ may also be expressed as a superposition of plane waves, an immediate consequence is the wave-point uncertainty relation, $\Delta_g x^i \Delta_g k_j \geq (1/2)\delta^i{}_j$. This is an uncertainty relation for delocalised 'points', not point-particles in the classical background of canonical QM [16]. A key question we must then address is, what is the momentum of a quantum geometry wave? For matter waves, $\mathbf{p} = \hbar \mathbf{k}$, but we have no *a priori* reason to believe that space must be quantised on the same scale as material bodies. In fact, setting $\Delta_g x \simeq l_{\rm Pl}$ and $\mathbf{p} = \hbar \mathbf{k}$ yields $\Delta_g p \simeq m_{\rm Pl} c$, giving a vacuum density of order $\rho_{\rm vac} \simeq (\Delta_g p)/(\Delta_g x)^3 c \simeq c^5/(\hbar G^2)$. This is essentially the same calculation as that given in Eq. (2), which results from the same physical assumptions. Hence, we set

$$\Delta_g x^i \Delta_g p_j \geq (\beta/2)\delta^i{}_j, \tag{4}$$

where $\beta \neq \hbar$ is the fundamental quantum of action for geometry. [2]

---

[2] In the relativistic regime, the tensor nature of gravitational waves must also be accounted for, but this may be neglected in the non-relativistic limit in which Eq. (4) remains valid [16]. In this model, a function is associated to each spatial point by doubling the degrees of freedom in the classical phase space and the classical point labeled by $\mathbf{x}$ is associated with the quantum probability amplitude $g(\mathbf{x} - \mathbf{x}')$. This is the mathematical representation of a delocalized 'point' in the quantum nonlocal geometry. For each $\mathbf{x}$, the additional variable $\mathbf{x}'$ may take any value in $\mathbb{R}^3$. Together, $\mathbf{x}$ and $\mathbf{x}'$ cover $\mathbb{R}^3 \times \mathbb{R}^3$, which is interpreted as a superposition of 3D Euclidean spaces [16]. The process of 'smearing' points is easiest to visualize in the case of a toy one-dimensional universe. In this case, the original classical geometry is the $x$-axis and the $(x, x')$ plane on which $g(x - x')$ is defined represents the smeared superposition of geometries. These issues are considered in detail in the refs. [16–19] (see, in particular, see Fig.

Smearing each point in the background convolves the canonical probability density with a Planck-width Gaussian. The resulting total uncertainties are

$$\Delta_\Psi X^i = \sqrt{(\Delta_\psi x^i)^2 + (\Delta_g x^i)^2}, \quad \Delta_\Psi P_j = \sqrt{(\Delta_\psi p_j)^2 + (\Delta_g p_j)^2}, \tag{5}$$

for each $i, j \in \{1, 2, 3\}$, where $\Psi := \psi g$ denotes the composite wave function of a particle in smeared space [16–19]. [3] Finally, we note that the existence of a finite cosmological horizon implies a corresponding limit on the particle momentum, which may be satisfied by setting $\Delta_g p \simeq \hbar \sqrt{\Lambda/3}$. The resulting quantum of action for geometry is

$$\beta \simeq \hbar \sqrt{\frac{\rho_\Lambda}{\rho_{\mathrm{Pl}}}} \simeq \hbar \times 10^{-61}. \tag{6}$$

The new constant $\beta$ sets the Fourier transform scale for $g(\mathbf{x} - \mathbf{x}')$, whereas the matter component $\psi(\mathbf{x})$ transforms at $\hbar$ [16, 19]. [4] However, this does not violate the existing no-go theorems for the existence of multiple quantisation constants. These apply only to species of material particles [25], and still hold in the smeared-space theory, undisturbed by the quantisation of the background [19].

## 4 The vacuum energy, revisited

The introduction of a new quantisation scale for space radically alters our picture of the vacuum, including our naive estimate of the vacuum energy. This must be consistent with the generalised uncertainty relations (5). Expanding $\Delta_\Psi X^i$ with $\Delta_g x^i \simeq l_{\mathrm{Pl}}$ gives the generalised uncertainty principle (GUP) and expanding $\Delta_\Psi P_j$ with $\Delta_g p_j \simeq \hbar \sqrt{\Lambda/3}$ yields the extended uncertainty principle (EUP), previously considered in the quantum gravity literature [26, 27].

Equations (5) may also be combined with the HUP, which holds independently for $\psi$ [16, 19], to give two new uncertainty relations of the form $\Delta_\Psi X^i \Delta_\Psi P_j \geq \cdots \geq (\hbar + \beta)/2 \cdot \delta^i{}_j$. The central terms in each relation depend on either $\Delta_\psi x^i$ or $\Delta_\psi p_j$, exclusively. Minimising the product of the generalised uncertainties, $\Delta_\Psi X^i \Delta_\Psi P_j$, we obtain the following length and momentum scales:

$$(\Delta_\psi x)_{\mathrm{opt}} \simeq l_\Lambda := \sqrt{l_{\mathrm{Pl}} l_{\mathrm{dS}}} \simeq 0.1\,\mathrm{mm},$$
$$(\Delta_\psi p)_{\mathrm{opt}} \simeq m_\Lambda c := \sqrt{m_{\mathrm{Pl}} m_{\mathrm{dS}}}\, c \simeq 10^{-3}\,\mathrm{eV}/c, \tag{7}$$

1 of ref. [16]), but are not discussed at length in the present article for want of space. Note also that classical points are defined, where necessary, as in standard differential geometry. However, the model considered here is not based on classical points or on the fixed manifolds that form the mathematical basis of classical spacetimes. Instead, we associate each point in the classical background, labelled by $\mathbf{x}$, with a vector in a quantum Hilbert space, $|g_\mathbf{x}\rangle$. The associated wave function, $\langle \mathbf{x}'|g_\mathbf{x}\rangle = g(\mathbf{x} - \mathbf{x}')$, may be regarded as a Gaussian of width $\sigma_g \simeq l_{\mathrm{Pl}}$. This represents the quantum state of a delocalized 'point' in the quantum geometry, but this term is used here in an imprecise sense, only for illustration. (Hence the inverted commas.)

[3] Note that, here, space is 'smeared', not in the sense implied by non-commutative geometry [20–23], but in the way that a quantum reference frame is smeared with respect to its classical counterpart [24]. More specifically, the model presented in [16–19] represents a nontrivial two-parameter generalisation (including both $\hbar$ and $\beta$) of the QRF formalism of canonical quantum mechanics. This corresponds to the modified de Broglie relation, $\mathbf{p}' = \hbar\mathbf{k} + \beta(\mathbf{k}' - \mathbf{k})$ [16], where the noncanonical term may be interpreted, heuristically, as the additional momentum 'kick' induced by quantum fluctuations of the nonlocal geometry. As stressed later in the main body of the text, this kind of generalisation evades the well known no go theorems for multiple quantisation constants [25], which apply only to species of material particles.

[4] The term 'quantum geometry wave', introduced above Eq. (4), therefore has a precise meaning. It refers to the plane wave components of $\tilde{g}_\beta(\mathbf{p} - \mathbf{p}')$, which is the $\beta$-scaled Fourier transform of $g(\mathbf{x} - \mathbf{x}')$. If $\sigma_g \simeq l_{\mathrm{Pl}}$ is the width of $g(\mathbf{x} - \mathbf{x}')$, the corresponding width of a delocalised point in momentum space is $\tilde{\sigma}_g \simeq \hbar\sqrt{\Lambda}$. The predictions of canonical quantum theory, in which quantum matter propagates on a sharp classical space(time) background, are recovered by taking the limits $\sigma_g \to 0$ and $\tilde{\sigma}_g \to 0$, simultaneously. Together, these yield $\beta \to 0$ [16].

where $m_{\mathrm{dS}} = \hbar/(l_{\mathrm{dS}}c) \simeq 10^{-66}$ g is the de Sitter mass. This gives a vacuum energy of order

$$\rho_{\mathrm{vac}} \simeq \frac{3}{4\pi} \frac{(\Delta_\psi p)_{\mathrm{opt}}}{(\Delta_\psi x)^3_{\mathrm{opt}} c} \simeq \rho_\Lambda = \frac{\Lambda c^2}{8\pi G} \simeq 10^{-30}\,\mathrm{g\,.\,cm^{-3}}\,, \tag{8}$$

as required. Taking $k_\Lambda = 2\pi/l_\Lambda$ as the UV cut off in Eq. (2), with $m = m_\Lambda$, also gives the correct order of magnitude value, $\rho_{\mathrm{vac}} \simeq \rho_\Lambda$ [16].

In this model, vacuum modes seek to optimise the generalised uncertainty relations induced by both $\hbar$ and $\beta$, yielding the observed vacuum energy. Any attempt to excite higher-order modes leads to increased pair-production of neutral dark energy particles, of mass $m_\Lambda \simeq 10^{-3}\,\mathrm{eV}/c^2$, together with the concomitant expansion of space required to accommodate them, rather than an increase in energy density [19]. The vacuum energy remains approximately constant over large distances, but exhibits granularity on scales of order $l_\Lambda \simeq 0.1$ mm [16, 28, 29]. It is therefore intriguing that tentative evidence for small oscillations in the gravitational force, with approximately this wavelength, has already been observed [30, 31].

## 5   Summary

The simple analysis above shows that, if space-time points are delocalised at the Planck length, $\Delta x \simeq l_{\mathrm{Pl}}$, the associated momentum uncertainty cannot be of the order of the Planck momentum, $\Delta p \neq \hbar/\Delta x \simeq m_{\mathrm{Pl}} c$. We are then prompted to ask: is it reasonable to assume that quantised waves of space-time carry the same quanta of momentum as matter waves with the same frequency? Though a common assumption, underlying virtually all attempts to quantise gravity that utilise a single action scale, $\hbar$, we note that it has, *a priori*, no theoretical justification. We have shown that relaxing this stringent requirement by introducing a new quantum of action for geometry, $\beta \neq \hbar$, leads to interesting possibilities, with the potential to open up brand new avenues in quantum gravity research [19, 32]. These include the proposal that the observed vacuum energy, and the present-day accelerated expansion of the universe that it drives, are related to the quantum properties of space-time [17, 18]. In this model, a measurement of the dark energy density constitutes a de facto measurement of the geometry quantisation scale, $\beta$, fixing its value to $\beta \simeq \hbar \times 10^{-61}$.

This essay was written as a non-technical introduction to the smeared-space model, whose formalism was developed over a series of published works [16–19, 32]. It is based on the material presented at the 4th International Conference on Holography, Hanoi, Vietnam (August 2020), and designed to be accessible to a wide and diverse audience. Interested readers are referred to the previous works [16, 18], in which the formalism was derived from the physical assumptions introduced above, and [19], which contains the most comprehensive summary of existing results, for full mathematical details. However, since these papers are long and complicated, a more technical, but still brief, introduction to the smeared-space theory is given in the Appendix.

## Acknowledgements

This work was supported by the Guangdong Province Natural Science Foundation, grant no. 008120251030. I am extremely grateful to Marek Miller and Shi-Dong Liang, for helpful comments and suggestions, and to Michael Hall, for bringing several references to my attention. Thanks also to the National Astronomical Research Institute of Thailand (NARIT) and the Research Center for Quantum Technology (RCQT), Chiang Mai University, for gracious hospitality during the final preparation of the manuscript.

# A   Details of the model

In [16], the smeared-space model quantum geometry was proposed, in which each point $\mathbf{x}$ in the classical background is associated with a vector in a Hilbert space,

$$|g_{\mathbf{x}}\rangle = \int g(\mathbf{x}' - \mathbf{x}) |\mathbf{x}'\rangle \, \mathrm{d}^3 \mathbf{x}', \tag{9}$$

where $\langle g_{\mathbf{x}} | g_{\mathbf{x}} \rangle = 1$. This describes a form of nonlocal geometry that is intrinsically quantum in nature, so that the width of $|g(\mathbf{x}' - \mathbf{x})|^2$ is assumed to be of the order of the Planck length [16, 18, 19], in accordance with our expectations from gedanken experiment arguments [33, 34].

It has long been known that classical nonlocal geometries, such as those introduced in [35], can be generated by first identifying each point in the classical manifold with a Dirac delta, $\delta^3(\mathbf{x} - \mathbf{x}')$. Nonlocality is then introduced by smearing each delta into a finite-width probability distribution $P(\mathbf{x} - \mathbf{x}')$, for example, a normalised Gaussian [36]. In this case, no new degrees of freedom are introduced, beyond those present in canonical quantum mechanics, since $\mathbf{x}'$ is simply a parameter that determines the position of $P$.

The smeared space model introduced in [16, 18] is different in that it first associates each point $\mathbf{x}'$ with a rigged basis vector of a Hilbert space, $|\mathbf{x}'\rangle$. The latter is then smeared to produce the normalised state (9). In this case, $\langle \mathbf{x}' | g_{\mathbf{x}} \rangle = g(\mathbf{x}' - \mathbf{x})$ is a genuine quantum mechanical amplitude, not a probability distribution. It has dimensions of $(\text{length})^{-3/2}$ not $(\text{length})^{-3}$ and, in principle, may contain nontrivial phase information. In this model, $|g_{\mathbf{x}}\rangle$ represents the state of a Planck-scale localised 'point' in the quantum geometry. Each Planck-scale localised point is then smeared into a superposition of all points in the background space by imposing the map

$$S : |\mathbf{x}\rangle \mapsto |\mathbf{x}\rangle \otimes |g_{\mathbf{x}}\rangle . \tag{10}$$

The smearing map (10) may be visualised as follows: for each point $\mathbf{x} \in \mathbb{R}^3$ in the classical geometry it generates one whole 'copy' of $\mathbb{R}^3$, thereby doubling the size of the classical phase space. The resulting smeared geometry is represented by a 6D volume in which each point $(\mathbf{x}, \mathbf{x}')$ is associated with a quantum probability amplitude, $g(\mathbf{x}' - \mathbf{x})$. This is interpreted as the amplitude for the coherent transition $\mathbf{x} \leftrightarrow \mathbf{x}'$ and the 6D phase space is interpreted as a superposition of 3D geometries [16, 18, 19].

In the nonrelativisitc limit, each geometry in the smeared superposition is Euclidean, but differs from all others by the pair-wise exchange of two points [18]. It is assumed that the interchange $\mathbf{x} \leftrightarrow \mathbf{x}'$ exchanges the canonical amplitudes, $\psi(\mathbf{x}) \leftrightarrow \psi(\mathbf{x}')$, which leads to additional fluctuations in the observed position of the particle, over and above those present in canonical quantum theory. We now review, briefly, how these fluctuations give rise to generalised uncertainty relations (GURs), including the GUP and EUP previously considered in the quantum gravity literature [26, 27].

For simplicity, we may imagine $|g(\mathbf{x}' - \mathbf{x})|^2$ as a normalised Gaussian centred on $\mathbf{x}' = \mathbf{x}$, but, here, $\mathbf{x}'$ is no longer a parameter. By introducing the tensor product structure (10) we have doubled the number of degrees of freedom of the theory, vis-à-vis canonical quantum mechanics. Those in the left-hand subspace, labelled by $\mathbf{x}$, represent the degrees of freedom of a canonical quantum particle, whereas those in the right-hand subspace, labelled by $\mathbf{x}'$, determine the influence of the background geometry. The action of $S$ on $|\mathbf{x}\rangle$ (10) then induces a map on the canonical quantum state, $|\psi\rangle = \int \psi(\mathbf{x}) |\mathbf{x}\rangle \, \mathrm{d}^3 \mathbf{x}$, such that

$$S : |\psi\rangle \mapsto |\Psi\rangle , \tag{11}$$

where

$$|\Psi\rangle = \int \int \psi(\mathbf{x}) g(\mathbf{x}' - \mathbf{x}) |\mathbf{x}, \mathbf{x}'\rangle \, \mathrm{d}^3 \mathbf{x} \, \mathrm{d}^3 \mathbf{x}'. \tag{12}$$

The square of the smeared-state wave function, $|\Psi(\mathbf{x},\mathbf{x}')|^2 = |\psi(\mathbf{x})|^2|g(\mathbf{x}'-\mathbf{x})|^2$, represents the probability distribution associated with a quantum particle propagating in the quantum geometry. Since $|\psi(\mathbf{x})|^2$ represents the probability of finding the particle at the fixed classical point $\mathbf{x}$ in canonical quantum mechanics, $|\psi(\mathbf{x})|^2|g(\mathbf{x}'-\mathbf{x})|^2$ represents the probability that it will now be found, instead, at a new point $\mathbf{x}'$. If $g(\mathbf{x})$ is a Gaussian centred on the origin, $\mathbf{x}' = \mathbf{x}$ remains the most likely value, but fluctuations within a volume of order $\sim \sigma_g^3$, where $\sigma_g$ is the standard deviation of $|g(\mathbf{x})|^2$, remain relatively likely [16, 18, 19]. Furthermore, since an observed position '$\mathbf{x}'$' cannot determine which point(s) underwent the transition $\mathbf{x} \leftrightarrow \mathbf{x}'$ in the smeared superposition of geometries, we must sum over all possibilities by integrating the joint probability distribution $|\Psi(\mathbf{x},\mathbf{x}')|^2$ over $d^3\mathrm{x}$, yielding

$$\frac{d^d P(\mathbf{x}'|\Psi)}{d\mathrm{x}'^3} = \int |\Psi(\mathbf{x},\mathbf{x}')|^2 d^3\mathrm{x} = (|\psi|^2 * |\mathbf{g}|^2)(\mathbf{x}'), \tag{13}$$

where the star denotes a convolution. In this formalism, only primed degrees of freedom represent measurable quantities, whereas unprimed degrees of freedom are physically inaccessible [16, 18, 19].

The variance of a convolution is equal to the sum of the variances of the individual functions, so that the probability distribution (13) gives rise to the GUR

$$(\Delta_\Psi X^i)^2 = (\Delta_\psi x'^i)^2 + (\Delta_g x'^i)^2. \tag{14}$$

This is the detailed derivation of the first of Eqs. (5), given in the main text. However, note that the primes on the physically measurable variables were omitted in Eqs. (5), for the sake of notational simplicity. It is straightforward to verify that (14) is obtained from the standard braket construction $(\Delta_\Psi X^i)^2 = \langle\Psi|(\hat{X}^i)^2|\Psi\rangle - \langle\Psi|\hat{X}^i|\Psi\rangle^2$, where

$$\hat{X}^i = \int x'^i d^3\hat{\mathcal{P}}_{\mathbf{x}'} = \hat{\mathbb{I}} \otimes \hat{x}'^i \tag{15}$$

is the generalised position-measurement operator and $d^3\hat{\mathcal{P}}_{\mathbf{x}'} = \hat{\mathbb{I}} \otimes |\mathbf{x}'\rangle\langle\mathbf{x}'| d^3\mathrm{x}'$ is the generalised projection.

Next, we note that the HUP, expressed here in terms of the physically accessible primed variables,

$$\Delta_\psi x'^i \Delta_\psi p'_j \geq \frac{\hbar}{2}\delta^i{}_j, \tag{16}$$

holds independently of Eq. (14). Combining the two and identifying the standard deviation of $|g(\mathbf{x})|^2$ with the Planck length according to [16],

$$\Delta_g x'^i = \sigma_g^i = \sqrt{2}l_{\mathrm{Pl}}, \tag{17}$$

then yields

$$\Delta_\Psi X^i \geq \frac{\hbar}{2\Delta_\psi p'_j}\delta^i{}_j\left[1 + \alpha(\Delta_\psi p'_j)^2\right], \tag{18}$$

where $\alpha = 4(m_{\mathrm{Pl}}c)^{-2}$, to first order in the expansion [16]. For $\Delta_\psi x'^i \gg \sigma_g^i \simeq l_{\mathrm{Pl}}$, we have that $\Delta_\Psi X^i \simeq \Delta_\psi x'^i$, and, in this limit, Eq. (18) reduces to the standard expression for the GUP [26].

In the momentum space picture, the composite matter-plus-geometry state $|\Psi\rangle$ is expanded as

$$|\Psi\rangle = \int\int \psi_\hbar(\mathbf{p})\tilde{g}_\beta(\mathbf{p}'-\mathbf{p})|\mathbf{p}\,\mathbf{p}'\rangle d^3\mathrm{p}\,d^3\mathrm{p}', \tag{19}$$

where

$$\tilde{\psi}_{\hbar}(\mathbf{p}) = \left(\frac{1}{\sqrt{2\pi\hbar}}\right)^3 \int \psi(\mathbf{x})e^{-\frac{i}{\hbar}\mathbf{p}.\mathbf{x}}d^3\mathbf{x},\tag{20}$$

as in canonical QM, and

$$\tilde{g}_{\beta}(\mathbf{p}'-\mathbf{p}) = \left(\frac{1}{\sqrt{2\pi\beta}}\right)^3 \int \mathbf{g}(\mathbf{x}'-\mathbf{x})e^{-\frac{i}{\beta}(\mathbf{p}'-\mathbf{p}).(\mathbf{x}'-\mathbf{x})}d^3\mathbf{x}',$$

where $\beta \neq \hbar$ is the fundamental quantum of action for geometry [16,18,19]. Note that, in Eq. (19), the basis $|\mathbf{p}\mathbf{p}'\rangle$ is entangled and cannot be separated into a simple tensor product state, i.e., $|\mathbf{p}\mathbf{p}'\rangle \neq |\mathbf{p}\rangle \otimes |\mathbf{p}'\rangle$. We emphasise this by not writing a comma in between $\mathbf{p}$ and $\mathbf{p}'$, by contrast with the position space basis, $|\mathbf{x}, \mathbf{x}'\rangle = |\mathbf{x}\rangle \otimes |\mathbf{x}'\rangle$. Nonetheless, $\tilde{g}_{\beta}(\mathbf{p}'-\mathbf{p})$ can be interpreted as the probability amplitude for the transition $\mathbf{p} \leftrightarrow \mathbf{p}'$ in smeared momentum space, by analogy with the position space representation [16]. A unitarily equivalent formalism, which is akin to a quantum reference frame transformation [24] of the formalism sketched here, but with $\hbar \leftrightarrow \beta$, and in which the position and momentum space bases are symmetrized, is presented in [18, 19].

The consistency of Eqs. (12) and (19) requires

$$\langle \mathbf{x}, \mathbf{x}'|\mathbf{p}\mathbf{p}'\rangle = \left(\frac{1}{2\pi\sqrt{\hbar\beta}}\right)^3 e^{\frac{i}{\hbar}\mathbf{p}.\mathbf{x}}e^{\frac{i}{\beta}(\mathbf{p}'-\mathbf{p}).(\mathbf{x}'-\mathbf{x})},\tag{21}$$

which is equivalent to the modified de Broglie relation

$$\mathbf{p}' = \hbar\mathbf{k} + \beta(\mathbf{k}'-\mathbf{k}).\tag{22}$$

This holds for particles propagating in the smeared background and the non-canonical term may be interpreted, heuristically, as an additional momentum 'kick' induced by quantum fluctuations of the spacetime [16, 18, 19]. We now fix $\beta$ from physical considerations and show how it is related to the minimum length and momentum scales of the GUP and EUP.

The general properties of the Fourier transform [4] ensure that the 'wave-point' uncertainty relation,

$$\Delta_g x'^i \Delta_g p'_j \geq \frac{\beta}{2}\delta^i{}_j,\tag{23}$$

holds in addition to Eq. (14) and the HUP (16), and that the inequality is saturated for Gaussian distributions. This is simply Eq. (4) from the main text, expressed more rigorously in terms of the requisite primed variables.

Next, we identify the standard deviation of $|\tilde{g}_{\beta}(\mathbf{p})|^2$ with the de Sitter momentum, which represents the minimum momentum of a particle whose de Broglie wave length is of the order of the radius of the Universe, $r_U \simeq l_{dS} = \sqrt{3/\Lambda}$,

$$\Delta_g p'_j = \tilde{\sigma}_{gj} = \frac{1}{2}m_{dS}c.\tag{24}$$

This yields the definition of $\beta$,

$$\beta := (2/3)\sigma^i_g\tilde{\sigma}_{gi} = (\sqrt{2}/3)l_{Pl}m_{dS}.\tag{25}$$

Written explicitly in terms of the observed dark energy density, Eq. (25) gives the value of $\beta$ obtained in Eq. (6) of the main text.

By analogous reasoning to that presented above, the probability of obtaining the observed value '$\mathbf{p}'$' from a smeared momentum measurement is

$$\frac{\mathrm{d}^3 P(\mathbf{p}'|\tilde{\mathbf{\Psi}})}{\mathrm{dp}'^3} = \int |\tilde{\Psi}(\mathbf{p}, \mathbf{p}')|^2 \mathrm{d}^3\mathrm{p} = (|\tilde{\psi}_{\hbar}|^2 * |\tilde{\mathbf{g}}_{\beta}|^2)(\mathbf{p}'), \tag{26}$$

which gives rise to the momentum space GUR

$$(\Delta_{\Psi} P_j)^2 = (\Delta_{\psi} p_j')^2 + (\Delta_g p_j')^2. \tag{27}$$

This is the second of Eqs. (5) from the main text, expressed in terms of primed variables, and can be obtained from the standard braket construction $(\Delta_{\Psi} P_j)^2 = \langle\Psi|(\hat{P}_i)^2|\Psi\rangle - \langle\Psi|\hat{P}_j|\Psi\rangle^2$ using

$$\hat{P}_j = \int p_j' \, \mathrm{d}^3 \hat{\mathcal{P}}_{\mathbf{p}'}, \tag{28}$$

where $\mathrm{d}^3\hat{\mathcal{P}}_{\mathbf{p}'} = \left(\int |\mathbf{p}\,\mathbf{p}'\rangle \langle\mathbf{p}\,\mathbf{p}'| \mathrm{d}^3\mathrm{p}\right)\mathrm{d}^3\mathrm{p}'$.

Substituting the HUP (16) into Eq. (27) and Taylor expanding to first order then yields

$$\Delta_{\Psi} P_j \geq \frac{\hbar}{2\Delta_{\psi} x'^i} \delta^i_{\ j} \left[1 + \eta(\Delta_{\psi} x'^i)^2\right], \tag{29}$$

where $\eta = (1/2)l_{\mathrm{dS}}^{-2}$ [16]. For $\Delta_{\psi} p_j' \gg \Delta_g p_j' \simeq m_{\mathrm{dS}}c$, we have $\Delta_{\Psi} P_j \simeq \Delta_{\psi} p_j'$ (27) and, in this limit, Eq. (29) reduces to the standard expression for the EUP.

Having obtained both the GUP and EUP from the smeared space formalism, we now show how they can be combined to give the so called extended generalised uncertainty principle (EGUP). This incorporates the effects of both canonical gravitational attraction and the presence of a constant background dark energy density on the microscopic dynamics of quantum particles [26, 27]. Combining Eqs. (14), (16) and (27), directly, gives

$$
\begin{aligned}
(\Delta_{\Psi} X^i)^2 (\Delta_{\Psi} P_j)^2 \;\geq\; & (\hbar/2)^2 (\delta^i_{\ j})^2 + (\Delta_g x'^i)^2 (\Delta_{\Psi} P_j)^2 \\
+\; & (\Delta_{\Psi} X^i)^2 (\Delta_g p_j')^2 \\
-\; & (\Delta_g x'^i)^2 (\Delta_g p_j')^2.
\end{aligned}
\tag{30}
$$

Substituting for $\Delta_g x'^i$ and $\Delta_g p_j'$ from Eqs. (17) and (24), taking the square root and expanding to first order, then ignoring the subdominant term of order $\sim l_{\mathrm{Pl}} m_{\mathrm{dS}} c$, yields

$$\Delta_{\Psi} X^i \Delta_{\Psi} P_j \geq \frac{\hbar}{2} \delta^i_{\ j} \left[1 + \alpha(\Delta_{\Psi} P_j)^2 + \eta(\Delta_{\Psi} X^i)^2\right]. \tag{31}$$

This is equivalent to the heuristic EGUP obtained in [27] but with $\Delta x^i$ and $\Delta p_j$ replaced by well defined standard deviations, $\Delta_{\Psi} X^i$ and $\Delta_{\Psi} P_j$. These represent the width of the composite matter-plus-geometry state $|\Psi\rangle$ in the position and momentum space representations, respectively [16, 18, 19].

Furthermore, it is possible to show that the product of generalised uncertainties, $\Delta_{\Psi} X^i \Delta_{\Psi} P_j$, is minimised when $\Delta_{\psi} x'^i$ and $\Delta_{\psi} p_j'$ take the values

$$(\Delta_{\psi} x'^i)_{\mathrm{opt}} = \sqrt{\frac{\hbar}{2} \frac{\Delta_g x'^i}{\Delta_g p_i'}}, \quad (\Delta_{\psi} p_j')_{\mathrm{opt}} = \sqrt{\frac{\hbar}{2} \frac{\Delta_g p_j'}{\Delta_g x'^j}}, \tag{32}$$

yielding

$$\Delta_{\Psi} X^i \, \Delta_{\Psi} P_j \;\geq\; \frac{(\hbar + \beta)}{2} \delta^i_{\ j}. \tag{33}$$

The same result is readily obtained from the Schrödinger–Robertson relation, $\Delta_\Psi O_1 \Delta_\Psi O_2 \geq (1/2)\langle\Psi|[\hat{O}_1,\hat{O}_2]|\Psi\rangle$, by noting that the commutator of the generalised position and momentum observables is

$$[\hat{X}^i,\hat{P}_j] = i(\hbar+\beta)\delta^i{}_j\hat{\mathbb{1}}. \tag{34}$$

The remaining commutators of the model are

$$[\hat{X}^i,\hat{X}^j] = 0, \quad [\hat{P}_i,\hat{P}_j] = 0. \tag{35}$$

Equations (34) and (35) show that GURs, including the GUP, EUP and EGUP, may be obtained without non-canonical modifications of the Heisenberg algebra [16, 18, 19]. (See also [37] for a similar result.) This allows the smeared space model to evade the problems that plague existing modified commutator models, including violation of the equivalence principle, the velocity-dependence of the minimum length, and the soccer ball problem [26, 38].

Finally, we note that substituting $\Delta_g x'^i = \sigma_g^i \simeq l_{\text{Pl}}$ and $\Delta_g p'_j = \tilde{\sigma}_{gj} \simeq m_{\text{dS}}c$ into Eqs. (32) yields the length and momentum scales given in Eqs. (7) of the main text, and, hence, the observed dark energy density over macroscopic distances. By contrast, using the standard HUP for the geometric part of the composite quantum wave function $\Psi = \psi g$, which is equivalent to taking the limit $m_{\text{dS}} \to m_{\text{Pl}}$, $\beta \to \hbar$ in the smeared-space model, yields the familiar 'worst prediction in theoretical physics', i.e., a vacuum energy of the order of the Planck density, $\rho_{\text{vac}} \simeq \rho_{\text{Pl}}$.

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
