# Peer review of "Why space could be quantised on a different scale to matter"

_SciPost Physics Proceedings, doi:SciPost Phys. Proc. 4, 014 (2021)_

## Round 1 · Referee Report · Anonymous (Referee 1) · 2020-10-19

Strengths
- The basic idea is interesting.
- applications related to the cosmological constant are given.
- Tries to address some major issues in Physics.
Weaknesses
- Important aspects are not clearly presented.
- No proposals to test the formalism are provided.
- There are mathematical issues that must be clarified.
Report
Requested changes
-
The meaning of the term "scale" is not clear. The quantization of matter takes place on the energy level; the quantization of gravity, if possible, would involve a geometrical scale. These two scales are not overlapping, and they are distinct, unless one introduces an energy scale for gravity. The meaning of "scale" in the present approach must be clarified.
-
How does the author defines a point?
-
Usually one can associate to a point the value of a function or of an operator, and not the function itself, so that in the point x_0 we may consider the couple (x_0,f(x_0)), where f is an arbitrary function or operator, but not the couple (x_0,f(x)), where x takes all real (or complex) values, let's say. The procedures of associating entire functions (operators), be they delta or of other type, to points, must be clearly explained.
-
The author also mention "geometry wave"(s), to which the rules of standard quantum mechanics are applied, without much justification. Using Eq. (2) for the "geometry wave"(s) is at least problematic. One must make clear what is the meaning of geometry wave in the present context, and why should they obey exactly the rules of quantum mechanics for matter.
-
The "quantum of action for geometry", given by Eq. (6), is an extremely small quantity, which looks very difficult to be measured directly, or even indirectly. Still the authors may suggest some other observational effects that could provide, at least in principle, some (realistic) observational signatures for $\beta$.
“Why space must be quantized on a different scale to matter” (scipost_202009_00001v1) - Revised Manuscript
Dear Editor(s),
Please see attached the revised manuscript of scipost_202009_00001v1. I have amended the text in order to address the referee’s comments and a detailed list of all changes is given below. These mostly take the form of footnotes, in order to preserve the essay-style flow of the original draft. I thank the referee for taking the time to read through the manuscript carefully and giving feedback. I hope that these changes have satisfactorily addressed his / her remaining concerns.
Please excuse me for taking so long to resubmit the paper, especially since the required changes were relatively minor, but I was caught up in unfortunate administrative duties these past few months.
With best wishes,
Matt
Changes:
• A footnote has been added at the end of the last sentence on pg 1 clarifying the meaning of the term ‘quantisation scale’.
• A footnote has been added at the end of the sentence below Eq. (3.1). This is to clarify the way in which whole functions, $g(x’-x)$, may be associated with single points, $x$, as well as how the model defines `points’ in both the quantum and classical regimes.
• A footnote was added at the end of sentence following Eq. (3.3). This is to clarify the precise meaning of the term ‘quantum geometry wave’.
• The final sentence of the Conclusions has been changed from “These include the proposal that the observed vacuum energy is related to the quantisation scale of space itself \cite{Lake:2019oaz,Lake:2019nmn}.” to “These include the proposal that the observed vacuum energy, and the present-day accelerated expansion of the universe that it drives, are related to the quantum properties of space-time \cite{Lake:2019oaz,Lake:2019nmn}. In this model, a measurement of the dark energy density constitutes a de facto measurement of the geometry quantisation scale, $\beta$, fixing its value to $\beta \simeq \hbar \times 10^{-61}$.” Although this is clearly a postdiction rather than a prediction of the model, it provides a concrete link between observations and the smeared-space model parameters, in response to the referee’s final point.
• Two new references have been added, which appear as refs. [5] and [23] in the new draft.
Attachment:
Why_space_must_be_quantised_on_a_different_scale_SciPost_v2.pdf
Reply to referee report for “Why space must be quantized on a different scale to matter” (scipost_202009_00001v1)
For clarity, we reproduce the referee’s comments in full, before replying to them point-by-point. Excerpts from the referee report are written in italics and our replies are written in normal type.
In the manuscript the author investigates the possibility that space-time is quantized on a different scale as compared to matter. The manuscript may be publishable in SciPost Physics Proceedings if the author would fully consider the following points:
1. The meaning of the term "scale" is not clear. The quantization of matter takes place on the energy level; the quantization of gravity, if possible, would involve a geometrical scale. These two scales are not overlapping, and they are distinct, unless one introduces an energy scale for gravity. The meaning of "scale" in the present approach must be clarified.
The 'scale’ referred to here is an action scale. In other words, the fundamental quantum of action for geometry in this model is $\beta = \sqrt{\rho_{\Lambda}/\rho_{\rm Pl}} \simeq 10^{-61} \hbar$, as in Eq. (3.3).
We note that the canonical quantisation of matter is also based on the adoption of a characteristic action scale, namely $\hbar$. Wave-particle duality is consistent with Poincare invariance in the relativistic limit and, hence, with Galilean invariance in the non-relativisitc limit, if and only if $\vec{p} \propto \vec{k}$ and $E \propto \omega$, as noted by Weinberg in his seminal work on QFT. Ultimately, it is the constant of proportionality, $\hbar$, that determines the length and momentum (energy) scales at which quantum effects become important for material bodies.
Similarly, in the model of nonlocal geometry considered here, $\beta << \hbar$ sets the length and momentum (energy) scales at which quantum effects become important for the background geometry, in which the canonical quantum matter propagates.
These issues are treated in detail in refs. [15], [17], [23] and [24] of the text. However, due to the limited time afforded to the conference talk on which this submission is based, and to the limited space afforded to the conference proceedings, it was not possible to elaborate on them more fully in this context.
2. How does the author defines a point?
Classical points are defined, where necessary, as in standard differential geometry. However, the model considered here is not based on classical points or on the fixed manifolds that form the mathematical basis of classical spacetimes.
Instead, we associate each point in the classical background, labeled by $\vec{x}$, with a vector in a Hilbert space, $\langle g_{\vec{x}}\rangle$. The associated wave function, $\langle\vec{x}|g_{\vec{x}}\rangle = g(\vec{x}’-\vec{x})$, may be regarded as a Gaussian of width $\sigma_g \simeq l_{\rm Pl}$. This represents the quantum state of a delocalized `point’ in the quantum geometry, but this term is used here in an imprecise sense, only for illustration (hence the inverted commas).
3. Usually one can associate to a point the value of a function or of an operator, and not the function itself, so that in the point x_0 we may consider the couple (x_0,f(x_0)), where f is an arbitrary function or operator, but not the couple (x_0,f(x)), where x takes all real (or complex) values, let's say. The procedures of associating entire functions (operators), be they delta or of other type, to points, must be clearly explained.
In this model, a function is associated to each point by doubling the degrees of freedom in the classical phase space. Hence, the classical point labeled by $\vec{x}$ is associated with the quantum probability amplitude, $g(\vec{x}’-\vec{x})$. This is the mathematical representation of a `delocalized point’ in the nonlocal geometry, as discussed above.
For each $\vec{x}$, the additional variable $\vec{x}’$ may take any value in $\mathbb{R}^3$. Together, $\vec{x}$ and $\vec{x}’$ cover $\mathbb{R}^6$, which is interpreted as a superposition of 3D Euclidean spaces.
However, the smearing process is easiest to visualize in the case of a toy one-dimensional universe. In this case, the original classical geometry is the x-axis and the $(x,x’)$ plane on which the function $g(x’-x)$ is defined represents the smeared superposition of geometries.
Again, these issues are considered in detail in the refs. [15], [17], [23] and [24], but are not discussed at length in the present article for want of space. (In particular, see Fig. 1 of ref. [15].)
4. The author also mention "geometry wave"(s), to which the rules of standard quantum mechanics are applied, without much justification. Using Eq. (2) for the "geometry wave"(s) is at least problematic. One must make clear what is the meaning of geometry wave in the present context, and why should they obey exactly the rules of quantum mechanics for matter.
Eq. (2.1) follows directly from the standard properties of the Fourier transform, which is applied to the function $g(\vec{x}’-\vec{x})$ at the scale $\beta$, just as one applies the Fourier transform to the function $\psi(\vec{x})$ at the scale $\hbar$ in canonical QM, in order to obtain the momentum space representation of the theory.
In this sense, delocalized spatial 'points' in our model are analogous to delocalized point-particles (i.e., wave functions) in canonical QM. However, the former are not assumed to follow the rules of standard quantum theory per se. There are a number of subtle differences between the quantum treatments of matter and geometry, which is what allows the model to evade the standard no go theorems regarding multiple quantisation constants [16].
The meaning of 'geometry wave’, here, is none other than this: that the 'wave function’ of a delocalized spatial point, $g(\vec{x}’-\vec{x})$, can be expanded in terms of both Dirac deltas and plane waves. The latter are analogous to the plane waves used to expand the wave function $\psi(\vec{x})$ in canonical QM, but with the canonical quantisation scale $\hbar$ replaced by $\beta$.
5. The "quantum of action for geometry", given by Eq. (6), is an extremely small quantity, which looks very difficult to be measured directly, or even indirectly. Still the authors may suggest some other observational effects that could provide, at least in principle, some (realistic) observational signatures for β.
If this model is correct, then the presence of a vacuum energy density of order $\rho_{\Lambda} = \Lambda c^2/(8\pi G)$ is already an observational signature of the geometry quantisation scale $\beta$. This is discussed briefly in Sec. 4, but is clearly of post-diction, not a prediction, of the model.
However, that is not all. As also discussed in Sec. 4, the model suggests that the true dark energy field optimizes the generalized uncertainty relations (3.2), which leads to granularity (i.e., small-scale fluctuations in the effective strength of the gravitational field) on scales of order 0.1 mm. At present, there is tentative (2$\sigma$) evidence that such fluctuations have already been observed [21,22], but more data is needed to either confirm or rule out this possibility.

Anonymous on 2020-10-19 [id 1011]
In the manuscript the author investigates the possibility that space-time is quantized on a different scale as compared to matter. The manuscript may be publishable in SciPost Physics Proceedings if the author would fully consider the following points:
The meaning of the term "scale" is not clear. The quantization of matter takes place on the energy level; the quantization of gravity, if possible, would involve a geometrical scale. These two scales are not overlapping, and they are distinct, unless one introduces an energy scale for gravity. The meaning of "scale" in the present approach must be clarified.
How does the author defines a point?
Usually one can associate to a point the value of a function or of an operator, and not the function itself, so that in the point x_0 we may consider the couple (x_0,f(x_0)), where f is an arbitrary function or operator, but not the couple (x_0,f(x)), where x takes all real (or complex) values, let's say. The procedures of associating entire functions (operators), be they delta or of other type, to points, must be clearly explained.
The author also mention "geometry wave"(s), to which the rules of standard quantum mechanics are applied, without much justification. Using Eq. (2) for the "geometry wave"(s) is at least problematic. One must make clear what is the meaning of geometry wave in the present context, and why should they obey exactly the rules of quantum mechanics for matter.
The "quantum of action for geometry", given by Eq. (6), is an extremely small quantity, which looks very difficult to be measured directly, or even indirectly. Still the authors may suggest some other observational effects that could provide, at least in principle, some (realistic) observational signatures for $\beta$.

---

## Round 2 · Referee Report · Anonymous (Referee 1) · 2021-2-10

Strengths

  1. New interpretation of the quantization process
  2. Development of a nonlocal quantum theoretical approach
  3. Cosmological applications, related to the interpretation of dark energy.

Weaknesses

  1. More physical applications and tests are necessary to support the main idea.

Report

The author has improved the initial version of the manuscript, and hence I think that teh present version is suitable for publication in SciPost.

---

## Round 2 · Referee Report · Anonymous (Referee 2) · 2021-3-29

Strengths

1- Deals with an important problem in theoretical physics

Weaknesses

1- The main claim does not logically follow from an argument as presented. 2- Some statements lack rigorous explanation. 3- Certain similar approaches in the literature are not mentioned. 4. Some important issues are not addressed.

Report

In the submitted manuscript the author suggests a new physical scale where geometry is quantized and proposes an uncertainty relation that involves this scale and explains the cosmological constant problem.

The idea of a quantization scale for spacetime which has no a priori relation to $\hbar$ and a possible coarse-graining of spacetime at high energy is certainly not new. One can find rigorous arguments and specific models for instance in Phys.Lett.B 331 (1994) 39-44 by Doplicher, Fredenhagen and Roberts or Annals Phys. 219 (1992) 187-198 by Madore. The author presents a similar suggestion, however it is not supported by rigorous arguments or a particular model that can realise it. The title of the manuscript claims that space must be quantised with a new scale, but the argumentation is that this must be so due to the smallness of the vacuum energy. It is not clear how the cosmological constant problem necessarily leads to this conclusion. In the body of the paper, the only explanation is that “we have no a priori reason to believe that space must be quantized on the same scale […]”. Although this is not incorrect, and indeed it has been suggested many decades ago, it does not follow as a logical conclusion from the cosmological constant problem.

Moreover, the lack of a specific model in the manuscript raises questions such as what is the precise mathematical description of the “smeared space”, how does classical geometry emerge, and whether Lorentz invariance is lost. None of these important issues is addressed. Based on the above, I cannot recommend the submitted manuscript for publication in its present form.

Requested changes

1- The main claim of the paper should be scaled down, already in the title and also in the main text. There is no rigorous argument establishing that a new quantization scale "must" exist. The proposal should be presented as a possibility, not as a certain conclusion.

2- Mention and comparison with previous similar proposals is required, such as the ones of Doplicher, Fredenhagen and Roberts, and Madore.

3- The author should explain how classical space(time) emerges, presumably in the limit $\beta \to 0$.

4- Previous approaches to the quantization of space at a new scale have addressed the question of doing this in a Lorentz invariant way. The author is asked to explain whether Lorentz invariance is lost or not, and if it does how does the proposal avoid being ruled out.

5- It would be advisable to support the proposal with a particular model/theory that can implement the modified uncertainty relation (4). Heisenberg's uncertainty relation leads to a noncommutative phase space; is there a similar noncommutative space or some other specific geometric picture that underlies the proposal of the paper?

  • validity: ok
  • significance: ok
  • originality: ok
  • clarity: ok
  • formatting: reasonable
  • grammar: perfect

Author:  Matthew J. Lake  on 2021-04-22  [id 1375]

(in reply to Report 2 on 2021-03-29)
Category:
reply to objection

Reply to the Referee Report for `Why space must be quantised on a different scale to matter' [scipost-202009-00001v2]

I thank the referee for their comments, and for drawing several references to my attention, which I was not previously aware of. I have cited these in the updated manuscript, together with some additional related literature. Below, I provide detailed replies to the points raised in the report. However, first, let me say that, as a contribution to a conference proceedings, my original draft was written with two important aims in mind:

  1. To provide an accurate written summary of the presentation I actually gave at the 4th International Conference on Holography in Hanoi, and

  2. To avoid undue technical detail in the text itself, which would be inappropriate for a proceedings article. Instead, I aimed to cite the relevant (already published) works, in which the technical discussion and mathematical details of the model are contained.

For these reasons, I have not made major changes to the manuscript. Instead, I address the questions raised by the referee, in depth, in this reply letter, and have added only short notes to the text to highlight the references where the relevant technical details can be found. For clarity, the referee's questions are given in quotation marks and my responses are written in normal type.

Report:

"In the submitted manuscript the author suggests a new physical scale where geometry is quantized and proposes an uncertainty relation that involves this scale and explains the cosmological constant problem.

"The idea of a quantization scale for spacetime which has no a priori relation to $\hbar$ and a possible coarse-graining of spacetime at high energy is certainly not new. One can find rigorous arguments and specific models for instance in Phys. Lett. B 331 (1994) 39-44 by Doplicher, Fredenhagen and Roberts or Annals Phys. 219 (1992) 187-198 by Madore."

I have read the works cited by the referee in detail, and cannot find any reference to a second quantisation scale. In the works by Madore and others, uncertainty relations for spatial coordinates are introduced by introducing noncommutative geometry (NCG) in the position space representation, e.g., by introducing an X-X commutator of the form $[X^i,X^j] = \sigma^2 \, \delta^{ij}$. Here, $\sigma$ is a constant with dimensions $[L]$ or, equivalently, $[M]^{-1}$ (if $\hbar = c = 1$). It is important to recognise that this does not constitute a second quantisation scale, which must have units of action, $[L][M]$. To obtain a new quantisation scale from NCG models, one must also introduce a similar relation in the momentum space representation, e.g., $[P_i,P_j] = \tilde{\sigma}^2 \, \delta_{ij}$, where $\tilde{\sigma}$ has dimensions $[M]$, or equivalently $[L]^{-1}$. It is then straightforward to see that, setting

$(\Delta X^i) _{min} \simeq \sigma$,

where $\sigma = l_{\rm Pl}$ is the Planck length, and

$(\Delta P_j)_{\rm min} \simeq \tilde{\sigma}$,

where $\tilde{\sigma} \simeq m_{\rm dS}$ and $m_{\rm dS} \simeq \hbar\sqrt{\Lambda}$ is the de Sitter mass, the new quantisation scale is $\beta \simeq \sigma \tilde{\sigma} \simeq \hbar\sqrt{\rho_{\Lambda}/\rho_{\rm Pl}}$. This is exactly Eq. (6) in the present text.

The studies by Doplicher, Fredenhagen and Roberts, and by Madore, cited above, certainly do not do this, and, to the best of my knowledge, neither do any of the subsequent related works. I am not sure why this is, and I confess that I am not an expert on NCG, but I could imagine that there are significant technical barriers to the consistent implementation of such a model.

However, if such a model were to be self-consistently constructed, it should be stressed that, since the X-X and P-P commutators above do not refer to the position and momentum of material particles, but, instead, to delocalised, or `smeared out' spatial points, the new quantisation scale $\beta \simeq \sigma \tilde{\sigma}$ need not have any a priori relation with $\hbar$. It would, instead, represent the quantisation scale for the spatial background on which canonical quantum matter propagates, which is much more reminiscent of our model than conventional NCG theories. Despite this, there are still many differences, which are discussed below in response to the referee's other questions.

"The author presents a similar suggestion, however it is not supported by rigorous arguments or a particular model that can realise it."

The formalism of the model is presented in a series of published works, which are cited at the relevant points in the manuscript, see refs. [16], [17], [18] and [19]. (Please note that the book chapter, [19], has not yet been formally published, because the hardback copy is still in print. However, the manuscript available on the arXiv has been accepted, in its current form, after a rigorous review process by Springer. I append the letter of acceptance to the end of this reply letter.)

"The title of the manuscript claims that space must be quantised with a new scale, but the argumentation is that this must be so due to the smallness of the vacuum energy. It is not clear how the cosmological constant problem necessarily leads to this conclusion."

The argument for this claim is based on the following, very general, observations:

  1. If the Planck length is a fundamental length scale in nature, then classical spatial points are in some way `delocalised' in the quantum theory of gravity. This introduces metric fluctuations over Planck-scale volumes and, hence, a minimum observable position $(\Delta X^i)_{\rm min}$ of the order of the Planck length. (This is certainly not a new idea and is a mainstay of most approaches to quantum gravity.)

  2. A Planck-scale fluctuation of the spacetime metric, over a volume $\sim (\Delta X)_{\rm min}^3$, must carry an associated momentum, which we label $\Delta P$. (The italicised word must here is important.)

  3. The associated energy density is of order $\rho \simeq \Delta P/(\Delta X)_{\rm min}^3$. We stress that this is the energy density induced by quantum fluctuations of the spacetime metric, i.e., the energy density of the quantum spatial background, not the energy density of the canonical quantum matter that propagates within it.

  4. On dimensional grounds, $\Delta P \simeq \kappa/l_{\rm Pl}$, where $\kappa$ has dimensions of action. Clearly, if $\kappa = \hbar$, i.e., if space is quantised on the same scale as matter, then $\rho \simeq \rho_{\rm Pl}$. Therefore, since the observed vacuum density is much lower than the Planck density, $\kappa \ll \hbar$.

Essentially, we argue that this conclusion is logically inherent in all previous works on quantum gravity (or at least those which assume Planck-scale metric fluctuations) but, for some reason, has never been fully explored in the literature. In Sections 3 and 4, we argue that setting $\kappa \equiv \beta \simeq \sigma\tilde{\sigma} \simeq \hbar\sqrt{\rho_{\Lambda}/\rho_{\rm Pl}}$, i.e., setting $\sigma \simeq l_{\rm Pl}$ and $\tilde{\sigma} \simeq m_{\rm dS}$, allows us to recover the observed vacuum energy density, $\rho_{\Lambda} \simeq \Lambda/G$, though it should be noted that, in order to realise this, we are required to make several other assumptions. The additional assumptions required are discussed, explicitly, in Section 4, but we make no strong claims for their acceptance.

As a happy byproduct, we also recover the GUP, EUP and EGUP, previously proposed in the quantum gravity literature. Equivalently, we may say that the generalised uncertainty relations naturally motivate a specific vacuum energy model, which saturates the EGUP, and that both arise, ultimately, from the existence of a second quantisation scale for space(time). Note that the EUP and EGUP, as presented in the existing literature, are recovered only when $\kappa \equiv \beta \propto \sqrt{\Lambda}$.

Nonetheless, the observation that $\rho_{\rm vac} \ll \rho_{\rm Pl}$, plus the existence of a Planck length cut-off for spatial wavelengths, requires $\kappa \ll \hbar$, regardless of whether this is identified with the observed value of the cosmological constant.

"In the body of the paper, the only explanation is that `we have no a priori reason to believe that space must be quantized on the same scale [...]' ”

As stated above, this is not the only explanation given in the body of the paper. The argument given in Sections 2 and 3 of the text was condensed into points 1-4 in the previous response.

"Although this is not incorrect, and indeed it has been suggested many decades ago, it does not follow as a logical conclusion from the cosmological constant problem."

I completely agree with this statement, but am not aware of any specific references in which a second quantisation scale (quantum of action) for space was suggested. If such a suggestion was made decades ago, then it is certainly an oversight on my part not to have cited these works! I would be most grateful if the referee could point me in their direction, and will also cite them in any future work on this topic.

"Moreover, the lack of a specific model in the manuscript raises questions such as what is the precise mathematical description of the “smeared space”, how does classical geometry emerge, and whether Lorentz invariance is lost. None of these important issues is addressed. Based on the above, I cannot recommend the submitted manuscript for publication in its present form."

The results quoted in the manuscript are based on a very specific model, with a rigorously defined mathematical formalism, which was developed in a series of papers, [16], [17], [18] and [19]. All the questions raised by the referee are explicitly addressed therein, e.g., how does classical geometry emerge?' (see ref. [16] Section 3.1, below Eq. (47), and Section 4.1),whether Lorentz invariance is lost?' (see ref. [19] Section 2). These points are discussed further below, but were not treated in detail in the present manuscript, since this would have been inappropriate for a conference proceedings.

(To be honest, I am not at all sure whether the referees were made aware, by SciPost, that the text is a contribution to a conference proceedings and was never intended as a research article. Therefore, it contains only a very brief summary of already published work. I stress that the omission of mathematical details is by design and that said details can be found in the works cited in the text. Needless to say, if the referee was misinformed, by SciPost, as to the nature and purpose of the article, this is in no way his / her fault.)

Requested changes:

"1. The main claim of the paper should be scaled down, already in the title and also in the main text. There is no rigorous argument establishing that a new quantization scale "must" exist. The proposal should be presented as a possibility, not as a certain conclusion."

To be honest, the title of the paper was not meant to be taken too literally. I completely agree with the referee, that such a title would be completely inappropriate for a research article, but I considered it within the scope of artistic license for the title of a conference talk. This was deliberately `provocative', to some degree, since it it was intended to grab, and hopefully keep, the attention of the audience. The title of the present manuscript is exactly the title of the talk I gave at the conference in Hanoi, because this is already a a matter of public record. (The conference program has long since been published.)

For this reason, with the referee's permission, I would like to keep the present title. However, I have no strong feelings on this either way, and, if he / she feels that another title would be more appropriate, I am happy to comply with this request. In this case, I would suggest either Why space could be quantised on a different scale to matter' orShould space be quantised on a different scale to matter?'.

Corresponding changes of language could also be made throughout the text, but I refer again to points 1-4 above, which I regard as a strong argument in favour of a new quantisation scale, $\kappa \ll \hbar$. A more detailed mathematical argument for its existence is given in [16], [18], [19]. See, for example, the treatment of delocalised (`smeared') momentum measurements, in a universe with a finite de Sitter horizon, given in [16], Section 3.1.3.

"2. Mention and comparison with previous similar proposals is required, such as the ones of Doplicher, Fredenhagen and Roberts, and Madore."

These have been added to the text. Once again, I thank the referee for bringing them to my attention.

"3. The author should explain how classical space(time) emerges, presumably in the limit $\beta \rightarrow 0$."

The emergence of the canonical quantum limit, that is, of quantum matter on a classical space(time) background, is dealt with explicitly in [16] (see Section 3.1, below Eq. (47)). The key point is that, although there are three ways to take the limit $\beta \rightarrow 0$, two of them lead to inconsistencies. Since $\beta \simeq \sigma\tilde{\sigma}$, where $\sigma$ sets the smearing scale for the position space representation and $\tilde{\sigma}$ sets the smearing scale for momentum space, setting $\beta \rightarrow 0$ by taking either $\sigma > 0$, $\tilde{\sigma} \rightarrow 0$ or $\sigma \rightarrow 0$, $\tilde{\sigma} > 0$ leads to a smearing of one representation while the other remains purely classical. In these cases, the mathematical formalism of the theory breaks down. Therefore, we are required to set $\sigma \rightarrow 0$ and $\tilde{\sigma} \rightarrow 0$, simultaneously. In this limit, we recover the predictions of canonical quantum theory [16]. A short note has been added to the text to highlight this point.

"4. Previous approaches to the quantization of space at a new scale have addressed the question of doing this in a Lorentz invariant way. The author is asked to explain whether Lorentz invariance is lost or not, and if it does how does the proposal avoid being ruled out."

This is an important point, which relates to one of the model's strongest advantages. It is well known that mainstream approaches to GUP models, based on modified commutation relations (including those derived from NCG) suffer from severe pathologies. These include:

  1. Violation of the equivalence principle.

  2. Violation of Lorentz invariance in the relativistic limit.

  3. The reference frame dependence of the `minimum' length.

  4. The inability to construct sensible multi-particle states, known as the `soccer ball problem'.

Ultimately, all of these problems, including the breaking of Lorentz invariance, arise from the breaking of the shift-isometry subgroup of the Poincare group, which also forms a subgroup of the Galilean group in the nonrelativistic limit. (See [19] and references therein, including the reviews of GUP literature by Hossenfelder [Living Reviews in Relativity volume 16, Article number: 2 (2013)] and Tawfik and Diab [Int. J. Mod. Phys. D 23 (2014) 1430025].) A major advantage of our model is that it generates generalised uncertainty relations (GURs) without introducing modified commutators of the form suggested in the existing literature. Instead, the canonical X-P commutator is simply rescaled, such that $\hbar \rightarrow \hbar + \beta$. (See also Bishop et al, Phys. Lett. B, Volume 816, 10 May 2021, 136265, for work in a similar direction.) This preserves translation invariance, even in the presence of a minimum length, since the reference frame of the background space is quantised, but not discretised.

Therefore, although we have not constructed an explicitly Lorentz invariant extension of the model, it is clear that a major obstacle to its implementation, which exists in all virtually all other GUP models, has been removed by the careful construction of our model in the nonrelativisitic limit. I completely agree with the referee that this point is important, but I did not have time to address it adequately in my conference talk, even in the Q and A. Therefore, since it is dealt with at length in already published work, I regarded it as beyond the scope of the present summary.

" 5. It would be advisable to support the proposal with a particular model/theory that can implement the modified uncertainty relation (4). Heisenberg's uncertainty relation leads to a noncommutative phase space; is there a similar noncommutative space or some other specific geometric picture that underlies the proposal of the paper?"

As explained above, the model is very particular, and is based on a rigorous mathematical formalism that was developed in a series of already published works, [16], [17], [18] and [19]. Below, I outline some of its essential features, including its differences with, and similarities to, previous models in the quantum gravity literature. I hope that this will clear up any remaining confusion.

With regard to this point, there is an unfortunate confusion of terminology, which, however, we were unable to avoid, even with the help of a thesaurus. In the review by Madore [arXiv:gr-qc/9709002], he states that points are somehow smeared out' orfuzzy'. It is important to recognise that points in our model are delocalised (smeared), not in the sense of NCG, but in the way that a quantum reference frame (QRF) is delocalised, or smeared, with respect to its classical counterpart. (See, for example, the work by Giacomini, Castro-Ruiz and Brukner [Nature Communications volume 10, Article number: 494 (2019)].) Importantly, this allows us to derive GURs, incorporating the minimum length and momentum scales, $(\Delta X^i)_{\rm min}$

of the order of the Planck legnth and $(\Delta P_j)_{\rm min}$ of the order of the de Sitter mass, even in the presence of commuting coordinates, i.e.,

$[X^i,X^j] = 0$, $[P_i,P_j] = 0$

[16,18,19]. The underlying geometric picture is illustrated, heuristically, in Figure 1 of ref. [16], for a toy one-dimensional universe.

More specifically, our model represents a nontrivial two-parameter generalisation (including $\hbar$ and $\beta$) of the formalism derived by Giacomini et al. This leads to a nontrivial generalisation of the canonical de Broglie relation (Eq. (38) in [16]), which, however, remains consistent with a generalised Galiliean and/or Poincare invariance. This generalisation consists in transforming space(time) `points' into superpositions thereof, translations into superpositions of translations, and Galilean or Lorentz velocity boosts into superpositions of boosts, etc. (Note, also, that trivial two-parameter generalisations of the form $p = \hbar k \mapsto p = \hbar k, \, p' = \beta k'$, i.e., models that treat quantum spacetime like quantum matter, but with a different quantisation constant, are already ruled out by well known no go theorems. See [19] and references therein for further discussion.) Although it was arrived at independently, and via somewhat different arguments, it is straightforward to verify that the formalism of the smeared space model [16,18,19] reduces, in the limit $\beta \rightarrow \hbar$, to the QRF formalism published in Nature Communications.

Strengths:

"1. Deals with an important problem in theoretical physics."

Weaknesses:

"1. The main claim does not logically follow from an argument as presented."

I cannot agree with this statement. While there is, of course, reasonable doubt over the validity of any scientific model, the results presented in the manuscript are supported by a rigorous mathematical formalism, which was developed logically from its underlying assumptions in a series of published works, [16], [17]. [18] and [19]. (There is also a new work, submitted as an invited contribution to a special issue of Quantum Reports, which further develops the implications of the model for quantum information theory, see [Quantum Rep. 2021, 3(1), 196-227].)

  1. Some statements lack rigorous explanation.

I completely agree! However, I stress again that this was by design, not by omission, and that the rigorous explanations and arguments for the results presented in the manuscript are given in [16], [17], [18] and [19]. In total, these works comprise nearly 150 pages of material, so that many compromises had to be made when preparing the talk summary.

"3. Certain similar approaches in the literature are not mentioned."

These have been added, together with a brief note explaining their similarities, and important differences, with the model presented.

"4. Some important issues are not addressed."

I hope that the comments above have now addressed these issues, to the referee's satisfaction.

Attachment:

Response_to_2nd_Referee_Report_scipost_202009_00001v2_Letter_2AFUbZA.pdf

---

## Round 2 · Author Response

Dear Editor(s),

Please see attached the revised manuscript of scipost_202009_00001v1. I have amended the text in order to address the referee’s comments and a detailed list of all changes is given below. These mostly take the form of footnotes, in order to preserve the essay-style flow of the original draft. I thank the referee for taking the time to read through the manuscript carefully and giving feedback. I hope that these changes have satisfactorily addressed his / her remaining concerns.

Best wishes,

Matt

---

## Round 2 · List of Changes

A footnote has been added at the end of the last sentence on pg 1 clarifying the meaning of the term ‘quantisation scale’.
• A footnote has been added at the end of the sentence below Eq. (3.1). This is to clarify the way in which whole functions, $g(x’-x)$, may be associated with single points, $x$, as well as how the model defines `points’ in both the quantum and classical regimes.
• A footnote was added at the end of sentence following Eq. (3.3). This is to clarify the precise meaning of the term ‘quantum geometry wave’.
• The final sentence of the Conclusions has been changed from “These include the proposal that the observed vacuum energy is related to the quantisation scale of space itself \cite{Lake:2019oaz,Lake:2019nmn}.” to “These include the proposal that the observed vacuum energy, and the present-day accelerated expansion of the universe that it drives, are related to the quantum properties of space-time \cite{Lake:2019oaz,Lake:2019nmn}. In this model, a measurement of the dark energy density constitutes a de facto measurement of the geometry quantisation scale, $\beta$, fixing its value to $\beta \simeq \hbar \times 10^{-61}$.” Although this is clearly a postdiction rather than a prediction of the model, it provides a concrete link between observations and the smeared-space model parameters, in response to the referee’s final point.
• Two new references have been added, which appear as refs. [5] and [23] in the new draft.

---

## Round 3 · Referee Report · Anonymous (Referee 2) · 2021-5-15

Strengths

  1. Explanations on the points raised in the previous report have been provided in the response letter.

Weaknesses

  1. No significant changes have been made in the resubmitted manuscript.

Report

The author's detailed response letter is acknowledged. However, given that there have not been significant changes in the resubmitted manuscript, save two footnotes, the status of it remains essentially the same as in the previous review. My recommendation to the Editors is to offer once more the opportunity to the author to improve the manuscript in the directions listed below.

Regarding the comments and explanations in the response letter:

  • In noncommutative geometry typically the momenta are also noncommuting. This is true in most specific models and certainly a central aspect of the programme of Madore, as can be seen already in the book "An Introduction to Noncommutative Differential Geometry and its Physical Applications".

  • The logical steps taken by the author to argue for the new "scale" are now better explained in the response letter. It still appears that the main claim is made in too strong terms and it must be scaled down to a proposal rather than a certainty (also in the title). Many aspects of the proposal are hard to evaluate without more rigorous explanations about the notion of "smeared points" and "smeared space" and the emergence of spacetime and geometry, or a specific theory. The author states that 150 pages of material where such explanations are made cannot be written in this contribution, which is of course true, however the reader cannot also read these 150 pages just to understand basic aspects where the proposal is based, such as the above. It is asked that some higher degree of precision is aimed at. A specific model and a clear description of the mathematical notions, geometry and the limits mentioned in the response letter is due.

  • Points 4 and 5 of the previous report are answered in some detail in the response letter, to the extent that they can be addressed. Since, as far as I understand, there is no page limit in this contribution, I suggest that the author suitably implements these explanations in the manuscript.

Requested changes

1- The issues of Lorentz invariance and emergence of spacetime (and general relativity therefore) should be discussed in more detail in the manuscript, building on the explanations in the response letter.

2- The basic notions used, such as "smeared space", "points" and the like, should be defined in more precise terms, essentially expanding and making good sense of footnote 2 in the manuscript. A specific, preferably 3+1 dimensional, model/theory where the proposal is realized should be discussed.

---

## Round 3 · Author Response

Dear Editor(s),

Please see attached the revised version of the manuscript (v3), which includes the changes referred to in my earlier reply letter. For clarity, these are highlighted in blue text.

With best wishes,

Matt

---

## Round 3 · List of Changes

Footnotes added in blue text.

---

## Round 4 · Referee Report · Anonymous (Referee 2) · 2021-7-18

Report

The author has made an appreciable effort to improve the manuscript. The added appendix provides technical justification for arguments appearing in the main text. Statements now appear in more appropriate form. Given that this is a proceedings contribution, the manuscript can be accepted for publication in the proceedings volume of the 4th International Conference on Holography, Hanoi, Vietnam in its current form.

---

## Round 4 · Author Response

Dear Editor(s), Please see attached the revised version of the manuscript (v4). I found it very difficult to include the requested amount of technical detail within the essay style flow of the original draft. Therefore, I included it in a separate appendix. This may be regarded as a "half-way house" between the 5 page non-technical summary of the model given in the main text, which is suitable for a wide and diverse audience, such as that attending the Hanoi conference for which it was originally written, and the 150+ page derivation of the full mathematical formalism given in refs. [16], [18] and [19]. I hope that this satisfies the remaining concerns of the referee. For clarity, all changes from v3 of the manuscript are highlighted in blue. Best wishes - Matt

---

## Round 4 · List of Changes

Technical detail of the model has been added in a new appendix. Minor changes of language, in accordance with the referee's previous recommendations, have also been made within the main text. All changes from the previous version of the manuscript are highlighted in blue.

---

## Editorial Decision

published